# Safety of Gonadal Tissue-Derived Mesenchymal Stem Cell Therapy in Geriatric Dogs with Chronic Disease

**DOI:** 10.3390/ani14142134

**Published:** 2024-07-22

**Authors:** So-Young Jeung, Ju-Hyun An, Sung-Soo Kim, Hwa-Young Youn

**Affiliations:** 1VIP Animal Medical Center, Seoul 02830, Republic of Korea; jeungs1007@gmail.com (S.-Y.J.); ilovekh0@vipah.co.kr (S.-S.K.); 2Laboratory of Veterinary Internal Medicine, Department of Veterinary Clinical Science, College of Veterinary Medicine, Seoul National University, Seoul 08826, Republic of Korea; 3Laboratory of Veterinary Emergency and Critical Care, Department of Veterinary Clinical Science, College of Veterinary Medicine, Kangwon National University, Chuncheon-si 24341, Republic of Korea; anjuhyun@kangwon.ac.kr

**Keywords:** adverse effects, aged animals, canine, elderly companion animals, geriatric dogs, gonadal tissue, mesenchymal stem cell, regenerative medicine, safety

## Abstract

**Simple Summary:**

Simple Summary: Nineteen geriatric dogs (aged 8–20 years) with chronic diseases, including myxomatous mitral valve disease, chronic kidney disease, intervertebral disk disease, cognitive dysfunction syndrome, degenerative joint disease, osteoarthritis, and progressive retinal atrophy, were included in this retrospective study. They were divided into two groups: a control group treated with adipose tissue-derived mesenchymal stem cells (MSCs) (n = 9) and a treatment group receiving MSCs derived from gonadal tissue (n = 10). MSC therapies were administered intravenously at monthly intervals, with multiple frequencies, using allogeneic MSCs. Safety was evaluated through short-term and long-term physical exams, blood tests, imaging studies, and the monitoring of adverse events. No clinical adverse effects were observed in the dogs treated with gonadal tissue-derived MSCs.

**Abstract:**

Ensuring the safety of mesenchymal stem cell (MSC) therapy is a fundamental requirement in clinical practice. This study aimed to assess the safety of using gonadal tissue-derived MSCs (n = 10) compared to the commonly utilized adipose tissue-derived MSCs (n = 9) in geriatric dogs with chronic diseases. All participants received allogeneic MSC therapy, and no allergic reactions due to allogeneic cell immunogenicity were noted. Both groups showed no adverse changes in physical exams or hematological parameters before and after therapy. Importantly, there were no instances of tumor formation or growth post-treatment in either group. The findings demonstrated that dogs treated with gonadal tissue-derived MSCs experienced no clinical adverse effects. However, clinical adverse effects were reported in one case of adipose tissue-derived MSC therapy. Despite limitations in monitoring beyond one year and constraints due to a small and diverse patient group, this pioneering study validates the safe use of gonadal tissue-derived MSCs in aged companion animals. It underscores the potential of utilizing tissues from neutering procedures to advance regenerative medicine and expand cell banks and therapy options for companion animals.

## 1. Introduction

Recent scientific advances in mesenchymal stem cell biology have stimulated the development and application of novel stem-cell-based therapeutic strategies in numerous animal diseases. Among the various types of stem cell populations, mesenchymal stem cells (MSCs), a multipotent adult progenitor cell population that can proliferate and differentiate into adipogenic, osteogenic, and chondrogenic lineages, have been the most widely used stem cell type in veterinary patients due to their promising therapeutic efficacy and convenient harvest and isolation procedures [1]. Additionally, MSC therapy poses a relatively low risk of tumorigenic development and fewer ethical challenges compared to embryonic stem-cell-based therapies [2].

Since the first clinical application of bone-marrow-derived mesenchymal stem cell (MSC) therapy, which aimed to improve the treatment outcomes of damaged tendons of racehorses in 2003 [3], there have been remarkable advancements in MSC therapies for treatments of various diseases in different animal species [4]. To date, the most promising therapeutic potentials have been highlighted for the treatment of orthopedic disorders, such as tendon injuries in horses and osteoarthritis (OA) in dogs [5,6,7]. Other diseases for which MSC therapy holds promising therapeutic potential include renal, neurologic, cardiac, liver, dental, skin, and ophthalmic disorders, as elegantly discussed elsewhere [4]. In particular, mesenchymal stem cell therapy has been shown to be beneficial in aging and age-related diseases [8]. Positive therapeutic outcomes of MSC therapies provide great promise for more effective and safer management strategies for various diseases that currently have limited treatment options.

Using gonadal tissue for mesenchymal stem cell therapy allows for easy acquisition during neutering surgeries, eliminating the need for separate procedures to obtain tissues as sources of MSCs. Moreover, performing autologous cell banking during neutering may alleviate concerns regarding the immunogenic side effects associated with the use of allogeneic cells. This highlights the importance of gonadal tissue-derived mesenchymal stem cells, showing that discarded tissues can be repurposed into valuable materials for regenerative medicine.

Currently, there is a lack of research on the safety of using gonadal tissue-derived MSCs in elderly companion animals with age-related chronic diseases. Here, our aim is to summarize the application of gonadal tissue-derived mesenchymal stem cell therapy in an animal hospital and discuss safety concerns related to treatment progress in geriatric dogs with chronic disease.

## 2. Materials and Methods

### 2.1. Study Population

All medical records of patients treated with MSCs were reviewed retrospectively. Mesenchymal stem cells extracted from visceral fat and gonad tissues (ovaries and testes) were prepared for injection into animals, with the origin and potential side effects of the MSCs fully explained to the animal owners beforehand. During the data-collection period from 21 November 2020 to 4 March 2023, a total of 69 patients received MSC therapy. Among these, 19 elderly dogs with chronic diseases met the criteria for a retrospective study and analysis. In this study, all medical records of patients treated with MSCs at the VIP Animal Medical Center during the specified period were reviewed retrospectively. The review procedure was approved by the Institutional Animal Care and Use Committee (IACUC) of VIP Animal Medical Center (Protocol No. VIP-0006-SC).

#### Inclusion and Exclusion Criteria

Patients were selected based on similar age, disease, sex, and weight parameters. Mesenchymal stem cell therapy was excluded for patients with severe conditions such as tumors, cardiogenic pulmonary edema, sepsis, thromboembolism, or infectious diseases. Furthermore, patients who were unable to undergo short-term or long-term follow-up post-MSC therapy or who lacked the essential evaluation metrics required for this study were excluded. As a result, out of the initial 69 dogs, 50 were excluded, yielding a final sample size of 19 dogs.

In this study, geriatric patients aged 8 to 20 years with degenerative diseases or chronic conditions managed for more than three months were included. The conditions comprised myxomatous mitral valve disease (MMVD), chronic kidney disease (CKD), proteinuria, intervertebral disk disease (IVDD), cognitive dysfunction syndrome (CDS), degenerative joint disease (DJD), osteoarthritis (OA), cranial cruciate ligament rupture (CCLR) and medial patellar luxation (MPL), and progressive retinal atrophy (PRA).

Before deciding on MSC therapy, a general screening examination was conducted, including a physical examination, blood analysis, imaging examinations, urine tests, and other necessary tests to check for the presence of cancer. The VDI pre-stem cell panel test with cancer panel was recommended, and MSC therapy was performed on patients whose tumors had been excluded. In addition, we also assessed the overall health status of the patients and identified their underlying medical conditions.

To evaluate the safety outcomes, patients who received MSC therapy at intervals of less than one month, on three or more sessions, were targeted for a one-year monitoring period. For tumor evaluation, physical examinations, blood tests, and imaging examinations were conducted annually.

### 2.2. Tissue Collection and Cell Preparations

#### 2.2.1. Origin Tissue Collection from Donor

We obtained adipose or gonadal tissues (testes or ovaries) that were discarded during the neutering surgery procedures at VIP Animal Medical Center with the consent of the owners. Donors were healthy dogs under one year of age. They were thoroughly screened for infectious diseases and demonstrated no abnormal findings based on both blood and imaging examinations. The process was approved by the Institutional Animal Care and Use Committee (IACUC) of VIP Animal Medical Center (Protocol No. VIP-0004-SC).

#### 2.2.2. MSCs Preparation (Isolation, Culture, Harvest, Dilution)

##### Tissue Preparation and Cell Isolation

Donor-derived tissue (testis, ovary, or adipose tissue) was extracted from a sterile container using sterilized forceps and transferred to a tube containing 20 mL of Dulbecco’s phosphate-buffered saline (DPBS; Thermo Fisher Scientific, Waltham, MA, USA) for primary washing. The tissue was then transferred to a Petri dish, and blood vessels and impurities were removed as thoroughly as possible using sterilized scissors and forceps. The impurity-free tissue was transferred to a new tube containing 20 mL of DPBS for secondary washing. The washed tissue was then transferred to a new Petri dish and cut into pieces approximately 1–2 mm in size using sterilized scissors until it became juicy.

The finely chopped tissue was sprinkled with 9 mL of MEM-alpha medium (α-MEM; Sigma-Aldrich, St. Louis, MO, USA) and 90 µL of 1% collagenase diluted solution (Sigma-Aldrich, St. Louis, MO, USA) in a tissue dish. The minced tissue was then transferred to a 50 mL conical tube. The conical tube was vortexed every 20 min, and the enzymatic reaction was carried out for 2 h at 37 °C in a water bath. The solution containing the tissue was filtered through a 100 µm cell strainer (SPL Life Sciences Co., Pocheon-si, Gyeonggi-do, Korea), and the filtrate was collected in a new tube. The strainer (100 µm) was then washed twice with 5 mL of culture medium.

When preparing the stromal vascular fraction, the cell suspension was filtered through a 100 µm strainer followed by a 40 µm strainer. The filtered cell pellet was then centrifuged at 700× *g* for 10 min at room temperature. After the stromal vascular fraction (SVF) was obtained, it was washed with DPBS at least three times. For culturing cells, the supernatant was removed after centrifugation, and the cells were resuspended in 1 mL of culture medium. The resuspended cells were transferred to a 100 mm culture dish with 9 mL of culture medium and evenly distributed. The culture dish containing the cells was incubated at 37 °C, 5% CO_2_. After 4 h, the culture medium was changed once to remove impurities. The culture medium was changed daily until non-MSC impurities were removed and then changed every two days thereafter.

##### MSCs Subculture

Once the cell confluency reached 80% or greater, subculturing was initiated. The media were removed from the dish through suction, followed by a wash with DPBS and additional suction. A total of 1 mL of trypsin/EDTA (Thermo Fisher Scientific, MA, USA) was evenly distributed to cells in a 100 mm dish to ensure detachment reaction. The dish containing trypsin/EDTA was then incubated at 37 °C with 5% CO_2_ for 2 min. To promote detachment, the plate was gently tapped parallel to the surface and examined under a microscope to ensure complete cell detachment. Culture medium (4 mL) was added to the dish to deactivate the trypsin/EDTA reaction, and the cells were transferred to a 50 mL conical tube. Any remaining cells were collected through suction and transferred to the conical tube after the addition of 5 mL of DPBS to the dish. The conical tube containing cells was centrifuged at 1500 rpm and 4 °C for 5 min, and the supernatant was removed to leave the cell pellet. The cell pellet was then resuspended in 2 mL of fresh culture medium, and the cell count was determined using a hemocytometer. Finally, 1 × 10^6^ cells were seeded in a new 100 mm dish containing 10 mL of culture medium, and the dish was incubated at 37 °C with 5% CO_2_. This subculture process was repeated for passages 1–4.

##### Cell Harvest and Dilution

To confirm the proper confluency of cells, microscopy was employed. After suctioning the media, the dish was washed with 5 mL of DPBS. A total of 1 mL of trypsin/EDTA was then applied to the dish and evenly distributed to ensure adequate reaction. The dish was then incubated at 37 °C for 2 min. Following incubation, the dish was gently tapped parallel to the surface to facilitate the detachment of cells, which were then observed under a microscope to ensure proper detachment. Next, 4 mL of culture medium was added to the dish to deactivate the trypsin/EDTA reaction, and the cells were collected by gently transferring them to a conical tube. Any remaining cells in the dish were collected by adding 4 mL of DPBS and suctioning them into the conical tube. The collected cells were then centrifuged at 1500 rpm, 4 °C for 5 min, and the supernatant was removed, leaving only the cell pellet.

To perform an additional wash before injection into the patient, DPBS (5 mL) was added to resuspend the cells. Cell counting was performed using a hemocytometer, and Trypan blue (10 µL) was mixed with cells (10 µL) at a 1:1 ratio for counting. The DPBS and diluted cells were centrifuged again, and the supernatant was removed. The pellet was diluted in 0.9% normal saline at a rate of 10 mL/kg/h for 30 min based on the patient’s body weight. For cardiac patients, a lower dilution ratio was used. The diluted MSCs were stored at 4 °C in refrigerator until the patient was ready for treatment.

#### 2.2.3. Cell Characterization

Gonadal and adipose tissue derived-mesenchymal stem cells had spindle-shaped fibroblast-like morphology. To characterize MSCs, immunophenotyping was performed using flow cytometry (BD Accuri™ C6 Plus, BD Biosciences, Franklin Lakes, NJ, USA). The majority of cells expressed positive for CD 29 (FITC) and CD 44 (FITC), but a few cells were negative for CD90 (PE). FITC-conjugated CD29 (antibody clone MEM-101A, Invitrogen, MA, USA) and FITC-conjugated CD44 (antibody clone IM7, Invitrogen, Waltham, MA, USA) were used for canine MSCs. PE-conjugated CD90 (antibody clone YKIX337.217, Invitrogen, Waltham, MA, USA) was used for canine MSCs.

MSCs were differentiated using commercial kits for 2 weeks and identified by staining adipocytes (with Oil red O staining), osteocytes (with Alzarin Red S staining), and chondrocytes (with Alcian blue staining) (StemPro Adipogenesis, Osteogenesis, and Chondrogenesis Differentiation kits, Thermo Fisher Scientific, MA, USA).

### 2.3. Patient Preparation and MSC Therapy

Ten dogs received gonadal tissue-derived MSC therapy in the experimental group, while nine received adipose tissue-derived MSC injections in the control group. The guardians were provided with written information regarding the potential adverse effects and complications of MSC therapy, and they provided their consent. The consent form is included in the Appendix A.

The decision to prepare cells for either allogeneic or autologous transplantation was made in consultation with the owner. We determined the application route, injection frequency and interval, passage, and number of injected cells based on the patient’s condition and disease.

For culture of MSCs, an MSC therapy reservation was made at least 5 days in advance. A total of 30 min of fluid therapy was administered before and after the cell therapy on the day of treatment. The main types of fluids used were 0.9% normal saline, and the speed was maintained at 2.5–5 mL/kg/h.

Before allogeneic MSCs therapy, chlorpheniramine maleate 0.2 mg/kg, SC, once (Histamine, SAMU MEDIAN Co., Ltd., Seoul, Republic of Korea) was administered 30 min before treatment to prevent immune reactions.

MSC intravenous injection was performed for approximately 20–30 min at a rate of 0.5–1 mL/min or less, and the syringe was gently rolled to ensure that the cells were fully dispersed before injection.

Patients were monitored for over an hour for hypersensitivity reactions before returning home. The patient returned for evaluation of treatment response or side effects according to the monitoring schedules. In many cases, concurrent therapy was administered depending on the underlying disease.

### 2.4. Medical Records

The patients’ medical records were reviewed using an electronic charting program (E-friends, pnV Co., Ltd., Seoul, Republic of Korea). The data were analyzed by comparing pre- and post-one-year mesenchymal stem cell therapy. This involved assessing adverse changes in physiological parameters and assessing tumor formation, growth, and any side effects through physical exams, blood tests, and imaging studies.

### 2.5. Adverse Events

After MSC therapy, patients underwent a one-year monitoring period to assess the occurrence of adverse events. The adverse events were recorded on a scale ranging from 0 to 3, reflecting the severity. Mild adverse effects included transient fever, local pain, fatigue, and allergic reactions. Moderate adverse effects encompassed pulmonary edema, gastrointestinal issues such as nausea, vomiting, and diarrhea, along with respiratory distress, hypotension, and arrhythmias. Severe adverse effects extended to tumor formation, thromboembolism, and even fatalities (Table 1).

### 2.6. Assessment of MSC Therapy Response

Patient outcomes were evaluated according to the criteria established in Shah K’s study [9]. The treatment response was recorded as “Substantial” (1; excellent improvement), “Positive” (2; good improvement), “No change” (3; no improvement), or “Worse” (4; deterioration in the patients’ condition due to MSC therapy).

### 2.7. Statistical Analysis

Statistical analysis was performed using software GraphPad Prism Version 9 (GraphPad, Inc., La Jolla, CA, USA). Numerical data were presented as mean ± standard deviation. MSC therapy safety outcomes based on variables were analyzed using a two-way ANOVA. The mean of differences was followed by a 95% confidence interval. The *p*-values were indicated by * *p* < 0.05 and ** *p* < 0.01 and were considered statistically significant.

## 3. Results

### 3.1. MSC Characterization and Differentiation

Gonadal tissue-derived MSCs displayed a morphology resembling spindle-shaped, fibroblast-like cells (Figure 1). Canine gonadal MSCs were assessed for MSC-positive markers such as CD29, CD44, and CD90, with expression levels consistently exceeding 95% (Figure 2). These MSCs were successfully differentiated into chondrocytes, osteoblasts, and adipocytes, as confirmed by positive staining results (Figure 3).

### 3.2. Study Animals

The gonadal tissue-derived MSC group consisted of dogs from seven breeds, while the adipose tissue-derived MSC group included dogs from six breeds (Table 2). In the gonadal tissue-derived MSC group, the mean age was 11.40 ± 3.07 years (range, 8~17 years), with a median age of 10.5 years. The mean weight was 5.36 ± 3.35 kg (range, 2.53~11.8 kg), and the median weight was 4 kg. There were five neutered females and five neutered males (Table 3). In the adipose tissue-derived MSC group, the mean age was 13.67 ± 4.00 years (range, 9~20 years), with a median age of 15 years. The mean weight was 6.56 ± 7.44 kg (range, 1.65~27 kg), and the median weight was 4.00 kg. There were six neutered females and three neutered males (Table 3).

### 3.3. MSC Target Disease Category

In the gonadal tissue-derived MSC group, MSC therapy was administered for various medical conditions, including cardiovascular disease, renal disease, neurologic disease, and orthopedic disease. In cardiovascular disease, the primary target was myxomatous mitral valve disease (MMVD). Renal diseases primarily involved chronic kidney disease (CKD) and proteinuria. Neurological diseases included intervertebral disk disease (IVDD) and cognitive dysfunction syndrome (CDS). Orthopedic diseases encompassed degenerative joint disease (DJD), osteoarthritis (OA), cranial cruciate ligament rupture (CCLR), and medial patellar luxation (MPL). On the other hand, the adipose-derived MSC group was applied to renal, orthopedic, cardiovascular, neurologic, and ophthalmic diseases, including progressive retinal atrophy (PRA) (Table 4).

### 3.4. Classification of MSCs

#### 3.4.1. Distribution of MSC Sources

Of 19 dogs, 10 dogs were given gonadal tissue-derived MSC therapy (6 dogs from ovaries and 4 dogs from testes), and 9 patients underwent adipose tissue-derived MSC therapy (Table 3).

#### 3.4.2. Distribution of MSC Origin

All patients received allogeneic MSC infusions (Table 3).

### 3.5. The Elements of MSC Application

#### 3.5.1. Delivery Routes of MSCs

All patients received intravenous MSC infusions (Table 5).

#### 3.5.2. Dose and Passage of MSCs

In the gonadal tissue-derived MSC group, the average MSC count was 1.28 ± 0.87 × 10^6^ cells/kg, whereas in the adipose tissue-derived MSC group, it was 1.24 ± 0.33 × 10^6^ cells/kg. All patients were transplanted MSCs using passage 4 MSCs (Table 5).

#### 3.5.3. Sessions of MSCs Administration

In the gonadal tissue-derived MSC group, the mean number of treatments was 9.50 ± 3.35 sessions (range, 3~12), with a median of 12 sessions. Conversely, in the adipose tissue-derived MSC group, the mean number of treatments was 4.44 ± 2.75 (range, 3~12), with a median of 3 sessions (Table 5).

### 3.6. Adverse Effects of Gonadal Tissue-Derived MSC Therapy

#### 3.6.1. Clinical Parameters

Only one dog reported fatigue for 1 day after adipose tissue-derived MSC therapy. No other mild to severe adverse effects were reported in either group. No tumor formation or growth was observed (Table 6).

#### 3.6.2. Physical Parameters

Patients with gonadal tissue-derived MSCs had no significant changes in physical exams. No notable changes in weight, body temperature, or blood pressure were seen at 1, 3, and 12 months of monitoring in both group (Table 7).

#### 3.6.3. Hematological Parameter

Blood tests (CBC, serum chemistry, and electrolytes) were conducted at 1, 3, and 12 months to monitor patient adverse effects. In the gonadal tissue-derived MSC group, a significant change in total protein was observed at the 1-month monitoring compared to baseline (*p* < 0.05) (Figure 4). In the electrolyte evaluation, a significant increase in Ca^2+^ was observed at 1 and 3 months (*p* < 0.05) (Figure 5). However, these changes in total protein and Ca^2+^ remained within the reference range. No other differences were observed in serum chemistry, CBC, and electrolyte test results (Table 8). As a control group, the adipose tissue-derived MSC group showed no significant differences in blood tests (Table 9).

### 3.7. Therapeutic Outcomes of MSC Therapy

The patients were scored by veterinarians based on the criteria for MSC therapy outcomes. There were 11 cases of excellent improvement, 5 cases of good improvement, and 3 cases of no improvement (Appendix A).

## 4. Discussion

The advancement of veterinary medicine has extended the lives of companion animals, leading to a growing population of elderly companion animals, which has become a commonplace occurrence in modern society [10,11,12]. Age-related diseases are now acknowledged as a distinct medical domain, pushing for advancements in medicine beyond conventional treatment limitations. As a result, regenerative medicine is steadily expanding to cater to the aging population [13]. MSC therapy remains one of the most prominent fields in regenerative medicine, offering low side effects and systemic effects, making it a promising option for elderly patients with multiple diseases.

Mesenchymal stem cells operate on the fundamental principle of promoting cellular regeneration and repair, with capabilities for anti-inflammatory and immunomodulatory effects [14]. Approaching aging pathologically involves the interaction between chronic inflammation and resultant cellular dysfunction, perpetuating a cycle of aging [15]. Therefore, focusing on anti-inflammatory therapy is crucial for mechanisms of aging prevention and age-related disease treatment [16]. Therefore, we need to view the application of MSCs positively in the treatment of age-related diseases.

Human and mouse studies investigated the differences in gonadal tissue-derived MSCs compared to other MSCs. Human putative ovarian-MSCs demonstrated multi-lineage differentiation capacity, with lower expression of Aminopeptidase N (ANPEP/CD13) compared to BM-MSCs and dermal fibroblasts [17]. The reduced ANPEP expression attenuated tumor cell proliferation and migration [18]. Human testes-derived MSCs showed a longer lifespan and higher proliferation rate compared to bone marrow-derived MSCs, which may offer advantages in MSC therapy preparation and patient treatment [19]. Mouse testes-derived MSCs have been shown to reach therapeutically necessary levels with high cell yield and proliferation rates in early passages [20]. Therefore, the superior proliferation capacity of gonadal MSCs offers advantages in both preparation and therapy for patients.

Testes and ovaries, being gonadal tissues, have long been targets for medical intervention. Within the realm of MSC research, various types of MSCs, including those derived from bone marrow, adipose tissue, and placenta, have been employed to address conditions such as testicular degeneration and ovarian insufficiency [21,22]. However, studies utilizing testis and ovary as sources of MSCs are exceedingly rare. Research on gonadal tissue-derived MSCs is primarily limited to in vitro cell isolation, differentiation, and characterization in human and mouse studies. To date, no large-scale clinical studies involving these MSCs in actual patient populations have been reported. Therefore, our aim is to validate the suitability of gonadal tissue as a source of MSCs by assessing the safety of treatment in elderly canine patients. 

In this study, mesenchymal stem cells were derived from fat or gonadal tissue discarded during neutering surgery in dogs under one year of age at a veterinary hospital. According to previous studies, the injection of allogeneic MSCs into dogs has been widely reported [23]. Notably, studies comparing cell potency based on donor age have shown that MSCs harvested from older donors exhibit significantly reduced potency compared to those from younger donors [24,25]. Consequently, for this study, which targeted diseased dogs aged 8 years and older, all patients received allogeneic MSC therapy. 

The VIP treatment system for supplying MSCs to patients is well-established, providing MSCs from a variety of tissues, not limited to fat. Thus, a retrospective analysis was conducted to confirm the safety of such treatments.

The safety of MSC therapy is a crucial prerequisite. Human adipose MSCs, up to passage 12, show no genetic alterations. In SCID mice, intravenous injections of low, medium, and high doses of MSCs over 13 weeks revealed no toxicities or adverse reactions. Similarly, high-dose injections in nude mice for 26 weeks showed no tumorigenesis [26]. Prior studies have affirmed the absence of tumor-associated risks with mesenchymal stem cells [27].

Previous clinical studies in dogs have reported various adverse effects following MSC therapy, but these have primarily consisted of minor or transient side effects. First, localized injections resulted in adverse effects in several studies. In one dog, temporary exacerbation at the injection site was reported following multiple attempts at the intra-articular injection of autologous adipose-derived MSCs for hip joint arthritis [28]. Swelling was observed for several days in two of the joints that received local injection therapy with allogeneic adipose-derived MSCs and hyaluronic acid for elbow dysplasia and osteoarthritis [29]. In a study of allogeneic neonatal MSC injections for osteoarthritis in multiple joints, mild and immediate self-limiting inflammatory joint reactions were observed in 5 out of 22 joints injected [30]. On the other hand, reported cases of systemic signs include a MUO patient who received autologous BM MSC injection and experienced temporary hyperthermia 24 h post-injection [31].

There have been reports of adverse events with unclear associations with MSCs. In a study of autologous olfactory mucosal stem cell injection for acute spinal cord injury, 3 out of 23 treated animals and 1 out of 11 control animals died due to factors including suspected disseminated intravascular coagulation, urinary tract infection, and thoracic hemorrhage. However, the relationship between MSC treatment and death remained unclear based on postmortem examinations [32]. In a study on chronic Chagas cardiomyopathy, autologous bone marrow-derived MSCs were injected into the right and left coronary arteries [33]. During the procedure, one animal experienced malignant irreversible arrhythmia and died. An autopsy revealed air embolism and myocardial infarction. Another animal died 14 days after the procedure, although no abnormalities were found during the autopsy. Although no prior medical history was documented, the possibility of paroxysmal arrhythmias as the cause of death was considered [33]. The arrhythmias are characteristic of Chagas cardiomyopathy, and the association between MSC therapy and mortality was ambiguous [34].

In this study, adipose tissue-derived MSCs, which have been widely used, were selected as the control group. The aim was to evaluate the safety of gonadal tissue-derived MSC therapy by comparing it with adipose-derived MSCs. All patients in this study received allogeneic mesenchymal stem cell therapy, and no hypersensitivity reactions due to allogeneic cell immunogenicity were observed. There were no adverse changes beyond the normal reference range in physical examinations and hematological monitoring before and after MSC therapy in both groups. Clinical adverse effects were not observed in the gonadal tissue-derived MSC therapy group. However, in the adipose tissue-derived MSC group, one case of fatigue was reported for 1–2 days after treatment. Additionally, no instances of neoplasm occurrence or growth were observed within one year. Consequently, it can be concluded that gonadal tissue-derived MSCs are as safe for patient treatment as adipose tissue-derived MSCs.

Following gonadal tissue-derived MSC administration, statistically significant in-creases in Ca^2+^ levels were observed at 1 and 3 months, and in total protein levels at 1 month, all within the normal range. Previous studies have suggested a link between Ca^2+^ and MSCs [35,36]. Ca^2+^ is known to play a crucial role in the signaling, proliferation, self-renewal, and differentiation of MSCs [35,37,38]. Furthermore, in clinical therapy, MSCs migrate to damaged tissue or organs through a homing effect, leading to interactions between MSCs and the vascular endothelium of the target tissue [39]. Increased intracellular Ca^2+^ regulates migratory signaling (FAK/Rho GTPase), ultimately enhancing cell migration and thus the efficiency of MSC therapy [40]. However, there have been no studies confirming changes in Ca^2+^ when MSCs are administered to animal models or patients. Although this study does not provide insight into the mechanism, it suggests the need for further research to understand the implications of increased calcium levels after MSC therapy. More investigation is necessary to assess its impact on future treatments. This study noted an increase in total protein levels following gonadal tissue-derived MSC treatment. Total protein comprises albumin and globulin [41]. Dehydration can lead to elevated albumin levels, while in pathological states such as infection or chronic inflammation, the activation of the host inflammatory response can stimulate immunoglobulin production, leading to an increase in globulin levels [42]. The statistical significance of albumin and globulin levels was assessed after gonadal tissue-derived MSC therapy, but no significant differences were found between baseline, short-term, and long-term measurements. Therefore, further research is needed to investigate the underlying cause of the observed increase in total protein levels.

As a minimum effort to ensure safe and effective treatment, we have developed a precise treatment process. Establishing standardization in the administration of MSCs to companion animals is of paramount importance. Various conditions for standardizing MSC administration have been documented in previous studies [24,43,44,45,46,47]. Drawing upon these studies, we have implemented standardized protocols for MSC treatment, including criteria for the age and passage number of MSCs, the characteristics of tissue donors, and the dosage of MSC administrations. Furthermore, we have developed and strictly adhered to our own quality control tests and MSC therapy protocols.

This study has several limitations. The retrospective design limited our ability to precisely control for patient and control group conditions. Additionally, our capacity to monitor adverse effects was constrained to a one-year post-MSC therapy period within a small patient group. The broad spectrum of diseases targeted within a limited sample size also introduces constraints regarding disease specificity. Future prospective studies with larger sample sizes, longer follow-up periods, and more stringent controls are necessary to validate our findings and address these limitations.

Nevertheless, this represents a pioneering investigation in veterinary medicine to clinically employ gonadal tissue-derived MSCs. Building upon this foundation, it has been ascertained that gonadal tissue-derived MSCs can be safely administered to aged companion animals with chronic conditions.

## 5. Conclusions

Consequently, this research not only validates the utilization potential of tissues procured during neutering procedures as valuable assets in regenerative medicine but also forecasts significant contributions to the expansion of cell banks and MSC therapy medicine for companion animals.

## Figures and Tables

**Figure 1 animals-14-02134-f001:**
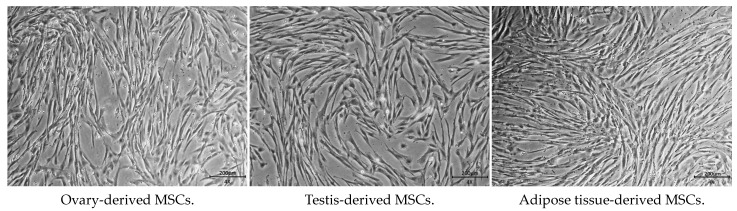
Morphology of gonadal and adipose tissue-derived mesenchymal stem cells. The morphology of canine gonadal and adipose tissue-derived MSCs observed under light microscopy reveals confluent cells exhibiting the typical spindle-like shape characteristic of MSCs, with a scale bar indicating 200 µm.

**Figure 2 animals-14-02134-f002:**
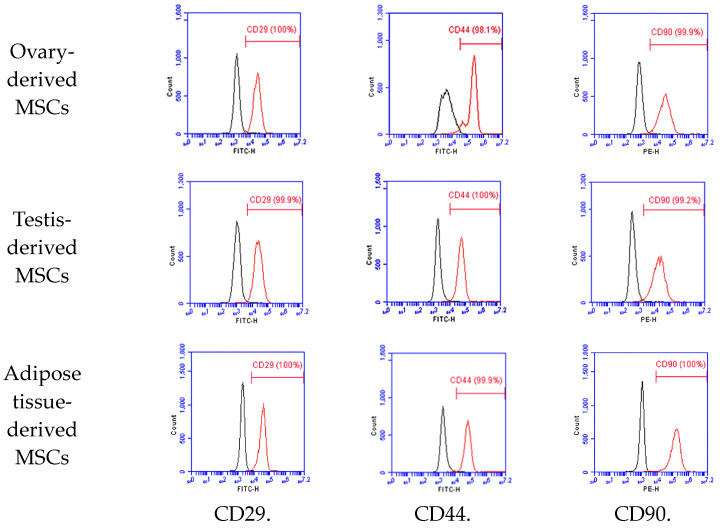
Cell surface marker expression of gonadal and adipose tissue-derived mesenchymal stem cells. The surface marker expressions of canine gonadal and adipose tissue-derived MSCs were depicted, with CD29, CD44, and CD90 expressions each exceeding 95%, as measured by flow cytometry.

**Figure 3 animals-14-02134-f003:**
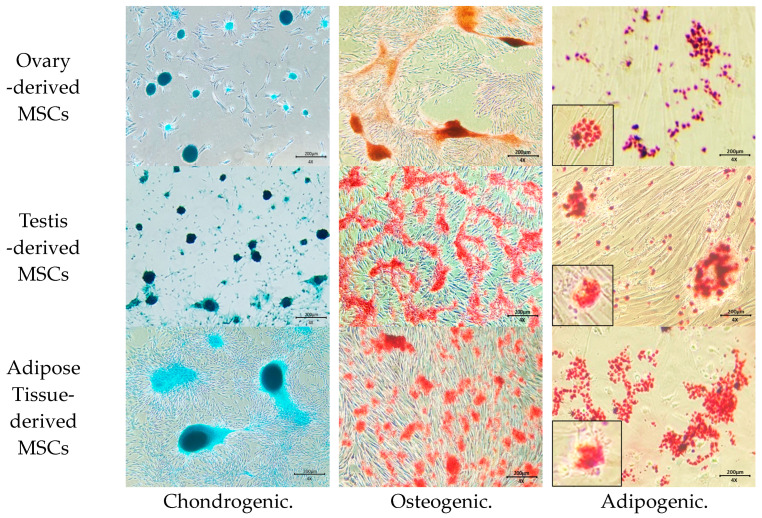
Differentiation of gonadal and adipose tissue-derived mesenchymal stem cells. Chondrocytes derived from MSCs were stained with Alcian blue, osteoblasts originating from MSCs were stained with Alizarin Red S, and adipocytes differentiated from MSCs were stained with Oil Red O. The differentiation of canine gonadal and adipose tissue-derived MSCs into chondrogenic, osteogenic, and adipogenic lineages was demonstrated.

**Figure 4 animals-14-02134-f004:**
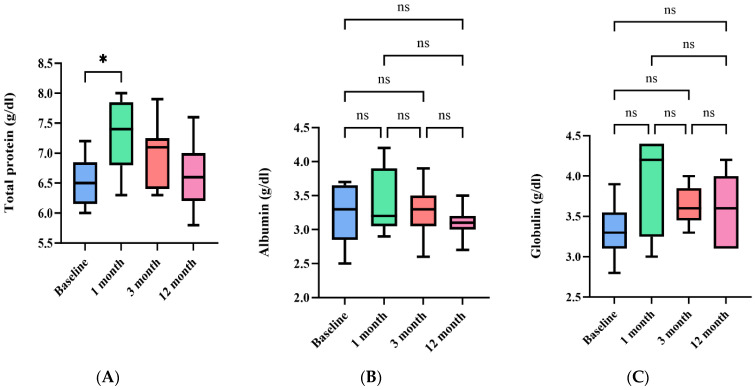
Periodic monitoring of proteins in the gonadal tissue-derived MSC group. Among (**A**) total protein, (**B**) albumin, and (**C**) globulin, total protein level exhibited a statistically significant increase at 1 month compared to baseline (*p* < 0.05). However, this change remained within the reference range. * *p* < 0.05, and ns indicates no significant differences.

**Figure 5 animals-14-02134-f005:**
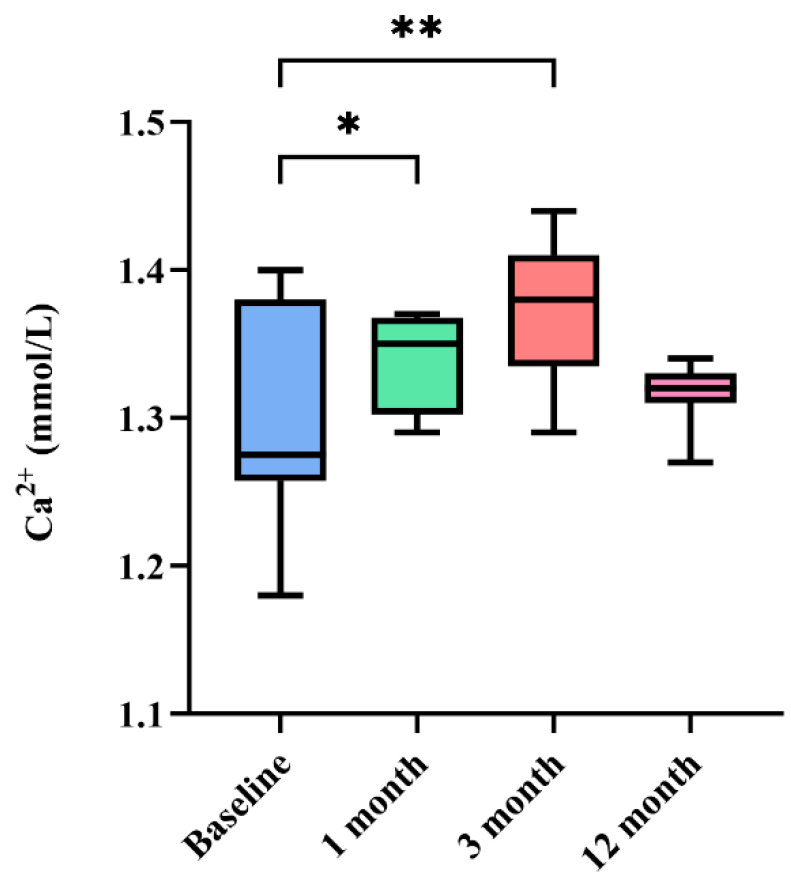
Periodic monitoring of ionized calcium in the gonadal tissue-derived MSC group. The ionized calcium level demonstrated a statistically significant increase at 1 and 3 months compared to baseline (*p* < 0.05). However, this change remained within the reference range. * *p* < 0.05, and ** *p* < 0.01.

**Table 1 animals-14-02134-t001:** Evaluation of adverse events in MSC therapy.

Grade	Assessment Characterization	Score
None	No adverse events	0
Mild	Transient fever, local pain, fatigue, and allergic reactions	1
Moderate	Pulmonary edema, gastrointestinal disturbances such as nausea, vomiting, and diarrhea, as well as respiratory distress, hypotension, and arrhythmias	2
Severe	Tumor formation, thromboembolism, death, and potentially fatal outcomes	3

**Table 2 animals-14-02134-t002:** Canine breed distribution in MSC therapy.

Variables	Gonadal Group		Adipose Group	
Breeds (N)	Beagle	1	Chihuahua	1
	Cocker Spaniel	1	Maltese	3
	Long coat Chihuahua	3	Mixed	1
	Maltese	2	Poodles	2
	Pomeranian	1	Weimaraner	1
	Poodles	1	Yorkshire Terrier	1
	Welsh Corgi	1	Chihuahua	1
Total	10		9	

**Table 3 animals-14-02134-t003:** Characteristics of canine MSC therapy recipients.

Variables	Gonadal Group	Adipose Group
Age (Mean ± SD years, range)	11.40 ± 3.07 years (range, 8~17)	13.67 ± 4.00 years (range, 9~20)
Median age	10.50 years	15 years
Body weight (Mean ± SD kg, range)	5.36 ± 3.35 kg (range, 2.53~11.8)	6.56 ± 7.44 kg (range, 1.65~27)
Median body weight	4.00 kg	4.00 kg
Sex (N)	Spayed female (5) Castrated male (5)	Spayed female (6) Castrated male (3)
MSC sources (N)	Ovaries (6) Testes (4)	Adipose (9)
MSC origin (N)	Allogeneic (10) Autologous (0)	Allogeneic (9) Autologous (0)

N = Patient number.

**Table 4 animals-14-02134-t004:** Classification of canine diseases with MSC therapy.

Variables	Gonadal Group			Adipose Group		
Disease category (N *)	Cardiovascular disease	6	MMVD	Renal disease	4	CKD
	Renal disease	4	CKD, Proteinuria	Orthopedic disease	3	DJD, OA
	Neurologic disease	2	IVDD, CDS	Cardiovascular disease	1	MMVD
	Orthopedic disease	1	DJD, OA, MPL, CCLR	Neurologic disease	2	IVDD, CDS
				Ophthalmic disease	1	PRA

* Includes patients with multiple diseases. MMVD = myxomatous mitral valve disease; CKD = chronic kidney disease; IVDD = intervertebral disk disease; CDS = cognitive dysfunction syndrome; DJD = degenerative joint disease; OA = osteoarthritis; MPL = medial patellar luxation; CCLR = cranial cruciate ligament rupture; PRA = progressive retinal atrophy.

**Table 5 animals-14-02134-t005:** Elements of MSC therapy.

Variables	Gonadal Group	Adipose Group
Dose (Mean ± SD, cells/kg)	1.28 ± 0.87 × 10^6^	1.24 ± 0.33 × 10^6^
Median dose (cells/kg)	1 × 10^6^	1.07 × 10^6^
Delivery route	Intravenous	Intravenous
Passage	P4	P4
Frequency (Mean ± SD, range, sessions)	9.50 ± 3.35 (range, 3~12)	4.44 ± 2.75 (range, 3~12)
Median frequency (sessions)	12	3

**Table 6 animals-14-02134-t006:** Clinical parameters of adverse effects after MSC therapy.

Variables	Gonadal Group	Adipose Group
None (Score 0)	10	8
Mild (Score 1)	0	1
Moderate (Score 2)	0	0
Severe (Score 3)	0	0
Total score	0	1

**Table 7 animals-14-02134-t007:** Physical parameters of adverse effects after MSC therapy.

Variable	Gonadal Group (N = 10)		Adipose Group (N = 9)		
Items	Mean ± SD (N)	*p*-Value between the Baseline and Each Point	Mean ± SD (N)	*p*-Value between the Baseline and Each Point	*p*-Value between Gonadal and Adipose Group
Bodyweight, kg					
Baseline	5.36 ± 3.54 (10)	-	6.56 ± 7.89 (9)	-	0.9873
1 month	5.37 ± 3.48 (10)	0.9990	6.65 ± 8.02 (9)	0.9743	0.9840
3 month	5.29 ± 3.58 (10)	0.8134	6.55 ± 7.75 (8)	0.8167	0.9914
12 month	4.86 ± 3.14 (8)	0.9950	4.10 ± 2.37 (5)	0.8633	0.9932
Temperature, °C					
Baseline	38.45 ± 0.46 (10)	-	38.48 ± 0.46 (9)	-	0.9999
1 month	38.36 ± 0.38 (10)	0.8974	38.19 ± 0.59 (9)	0.1575	0.8701
3 month	38.39 ± 0.40 (10)	0.9661	38.20 ± 0.48 (8)	0.4766	0.8376
12 month	38.31 ± 0.36 (8)	0.9012	38.38 ± 0.24 (5)	0.9643	0.9980
Systolic blood pressure, mmHg					
Baseline	141.30 ± 18.95 (10)	-	153.22 ± 10.88 (9)	-	0.3856
1 month	143.50 ± 14.15 (10)	0.9816	149.22 ± 24.80 (9)	0.8955	0.9054
3 month	138.50 ± 17.00 (10)	0.9634	144.75 ± 11.46 (8)	0.8911	0.8865
12 month	144.88 ± 13.15 (8)	0.9959	145.00 ± 10.00 (5)	0.7726	>0.9999

**Table 8 animals-14-02134-t008:** Periodic blood test monitoring of gonadal tissue-derived MSC group.

		Mean ± SD						*p*-Value		
Blood Analysis	Category	Baseline (N)	1 Month (N)	3 Month (N)	12 Month (N)	Reference Range	Unit	Baseline 1 Month	Baseline 3 Month	Baseline 12 Month
CBC	Hematocrit	45.17 ± 6.76 (10)	44.4 ± 5.80 (4)	44.36 ± 5.31 (10)	44.77 ± 6.20 (7)	37.3–61.7	%	0.9985	0.9467	0.8695
	WBC	8.02 ± 2.77 (10)	11.39 ± 6.44 (4)	8.93 ± 3.28 (10)	9.95 ± 3.45 (7)	5.05–16.76	K/µL	0.2415	0.8378	0.5619
	Neutrophils	5.61 ± 2.38 (10)	8.82 ± 6.11 (4)	6.13 ± 2.17 (10)	7.12 ± 3.07 (7)	2.9–11.64	K/µL	0.3226	0.9541	0.7208
	Lymphocytes	1.53 ± 0.57 (10)	1.8 ± 0.58 (4)	1.75 ± 0.71 (10)	1.68 ± 0.69 (7)	1.05–5.10	K/µL	0.0541	0.1381	0.7999
	Monocytes	0.42 ± 0.14 (10)	0.47 ± 0.16 (4)	0.48 ± 0.16 (10)	0.58 ± 0.16 (7)	0.16–1.12	K/µL	0.5322	0.7379	0.1291
	Platelets	359.3 ± 149.2 (10)	418.5 ± 120.4 (4)	360.9 ± 191.3 (10)	391.4 ± 147.1 (7)	148–484	K/µL	0.9919	>0.9999	0.8775
Serum chemistry	Glucose	105.4 ± 15.22 (9)	105.8 ± 14.91 (4)	97.67 ± 13.39 (9)	99.43 ± 14.48 (7)	60–120	mg/dL	0.9953	0.2242	0.1632
	BUN	18.83 ± 11.9 (10)	20.9 ± 10.48 (6)	24.88 ± 13.33 (10)	25.88 ± 20.01 (8)	7–27	mg/dL	0.9966	0.4453	0.1998
	Creatinine	1.19 ± 0.57 (10)	1.22 ± 0.21 (6)	1.24 ± 0.92 (10)	1.08 ± 0.57 (8)	0.5–1.8	mg/dL	0.7769	0.9846	0.8062
	ALP	185.7 ± 107.5 (9)	214.6 ± 128.4 (5)	145.1 ± 105.8 (9)	154.1 ± 60.23 (7)	23–212	U/L	>0.9999	0.3497	0.8657
	ALT	165.8 ± 237 (9)	196.2 ± 132.7 (5)	103.3 ± 54.95 (9)	113.3 ± 96.4 (7)	10–100	U/L	>0.9999	0.7767	0.7381
	Total protein	6.51 ± 0.41 (9)	7.34 ± 0.64 (5)	6.94 ± 0.52 (9)	6.66 ± 0.6 (7)	5.2–8.2	g/dL	0.039	0.1181	0.9855
	Albumin	3.21 ± 0.42 (9)	3.42 ± 0.5 (5)	3.28 ± 0.38 (9)	3.09 ± 0.24 (7)	2.2–3.9	g/dL	0.7978	0.9588	0.815
	Total bilirubin	0.3 ± 0.1 (5)	N/A	0.43 ± 0.28 (7)	0.28 ± 0.15 (4)	0.0–0.9	mg/dL	N/A	0.2367	0.9551
	GGT	6.83 ± 5.35 (6)	N/A	7.29 ± 7.72 (7)	3 ± 4.2 (6)	0–10	U/L	N/A	0.9439	0.7331
	Total cholesterol	188.2 ± 32.17 (6)	N/A	193.3 ± 45.92 (7)	187.8 ± 27.11 (5)	110–320	mg/dL	N/A	0.9966	0.9975
	Phosphorus	3.62 ± 0.96 (7)	3.45 ± 0.5 (2)	4.06 ± 1.92 (8)	3.88 ± 0.78 (6)	2.5–6.8	mg/dL	0.9285	0.56	0.619
	Amylase	876.2 ± 768.4 (6)	N/A	700.1 ± 442.4 (7)	1040 ± 774.5 (5)	226–1063	U/L	N/A	0.8887	0.8823
	Lipase	813.8 ± 126.2 (4)	N/A	658.3 ± 140.5 (7)	746.8 ± 254.1 (5)	200–1800	U/L	N/A	0.6039	0.1188
	AST	40.43 ± 20.48 (7)	44 ± 0 (1)	39.71 ± 14.47 (7)	39.67 ± 7.941 (6)	0–50	IU/I	0.6017	0.9934	0.9939
	Triglycerides	58.57 ± 21.97 (7)	N/A	92.86 ± 53.83 (7)	81.5 ± 27.7 (6)	10–100	mg/dL	N/A	0.3248	0.9872
Electrolytes	Na^+^	149.5 ± 3.6 (10)	148.5 ± 2.65 (4)	150.3 ± 1.5 (9)	151.1 ± 1.07 (7)	145–151	mmol/L	0.8261	0.6053	0.234
	K^+^	4.44 ± 0.43	4.14 ± 0.31	4.25 ± 0.49	4.23 ± 0.3	3.9–5.1	mmol/L	0.9922	0.2574	0.8142
	Ca^2+^	1.3 ± 0.07	1.34 ± 0.04	1.37 ± 0.05	1.32 ± 0.02	1.16–1.40	mmol/L	0.0408	0.0029	0.6557
	Cl^−^	115.6 ± 5.13	113 ± 9.27	114.3 ± 5.05	115.4 ± 2.64	110–119	mmol/L	0.7373	0.8902	0.8872

**Table 9 animals-14-02134-t009:** Periodic blood test monitoring of adipose tissue-derived MSC group.

		Mean ± SD						*p*-Value		
Blood Analysis	Category	Baseline (N)	1 Month (N)	3 Month (N)	12 Month (N)	Reference Range	Unit	Baseline 1 Month	Baseline 3 Month	Baseline 12 Month
CBC	Hematocrit	38.1 ± 6.18 (8)	34.94 ± 10.35 (7)	37.88 ± 6.33 (5)	41.58 ± 8.73 (5)	37.3–61.7	%	0.8496	>0.9999	0.8889
	WBC	10.06 ± 4.39 (8)	12.76 ± 9.12 (7)	11.92 ± 7.96 (5)	10.94 ± 2.07 (5)	5.05–16.76	K/µL	0.6965	0.6861	0.3765
	Neutrophils	6.63 ± 2.78 (8)	8.88 ± 6.95 (7)	8.21 ± 5.62 (5)	7.44 ± 2.73 (5)	2.9–11.64	K/µL	0.6604	0.5776	0.352
	Lymphocytes	1.81 ± 0.70 (8)	2.28 ± 1.66 (7)	1.70 ± 0.85 (5)	1.93 ± 1.21 (5)	1.05–5.10	K/µL	0.522	0.9959	0.8897
	Monocytes	1.17 ± 1.54 (8)	1.24 ± 1.25 (7)	1.73 ± 2.25 (5)	1.08 ± 0.31 (5)	0.16–1.12	K/µL	0.9997	0.5926	0.3974
	Platelets	423.1 ± 155.8 (8)	424.6 ± 151.5 (7)	611.4 ± 250.6 (5)	514.4 ± 238.1 (5)	148–484	K/µL	0.9967	0.1362	0.9306
Serum chemistry	Glucose	105.3 ± 11.59 (7)	97.8 ± 17.34 (5)	103 ± 13.32 (5)	106.6 ± 23.53 (5)	60–120	mg/dL	0.9984	0.9939	0.9584
	BUN	28.89 ± 14.68 (8)	40.13 ± 38.34 (7)	28.74 ± 12 (5)	36.08 ± 25.28 (5)	7–27	mg/dL	0.7318	0.9997	0.2581
	Creatinine	1.44 ± 1.22 (8)	1.437 ± 1.39 (7)	1.29 ± 0.86 (5)	1.03 ± 0.90 (5)	0.5–1.8	mg/dL	0.9276	0.9982	0.86
	ALP	156 ± 106.4 (7)	150.6 ± 94.74 (5)	179.6 ± 118.7 (5)	357 ± 300.3 (5)	23–212	U/L	0.9327	0.9405	0.115
	ALT	175.1 ± 144.3 (7)	95 ± 51.91 (6)	68.6 ± 47.05 (5)	157.6 ± 168.4 (5)	10–100	U/L	0.5806	0.6918	0.9998
	Total protein	7.01 ± 0.78 (7)	6.64 ± 0.81 (5)	6.6 ± 0.52 (5)	6.84 ± 1.47 (5)	5.2–8.2	g/dL	0.7865	0.6627	0.9444
	Albumin	3.24 ± 0.36 (7)	2.95 ± 0.38 (6)	2.88 ± 0.24 (5)	2.96 ± 0.55 (5)	2.2–3.9	g/dL	0.9251	0.3949	0.3004
	Total bilirubin	0.24 ± 0.08 (6)	0.88 ± 1.58 (5)	0.21 ± 0.08 (5)	0.31 ± 0.14 (4)	0.0–0.9	mg/dL	0.9004	0.733	0.991
	GGT	14.5 ± 20.56 (6)	12.2 ± 9.68 (5)	16.4 ± 17.29 (5)	39.75 ± 46.38 (4)	0–10	U/L	0.9276	0.9812	0.1444
	Total cholesterol	193.3 ± 13.02 (6)	178 ± 14 (5)	182.2 ± 17.75 (5)	249.8 ± 74.75 (4)	110–320	mg/dL	0.509	0.9348	0.4823
	Phosphorus	5.46 ± 0.84 (7)	5.17 ± 1.38 (6)	4.98 ± 0.77 (5)	5.64 ± 1.01 (5)	2.5–6.8	mg/dL	0.9168	0.6009	0.191
	Amylase	1195 ± 726.5 (6)	1542 ± 851.3 (5)	1554 ± 642.3 (5)	1463 ± 1042 (4)	226–1063	U/L	0.8725	0.8249	0.0631
	Lipase	2096 ± 1450 (5)	3266 ± 1652 (4)	2093 ± 2134 (4)	697 ± 261.6 (2)	200–1800	U/L	>0.9999	0.9962	0.9781
	AST	39.33 ± 17 (6)	40.25 ± 20.56 (4)	31.75 ± 7.411 (4)	44.5 ± 29.89 (4)	0–50	IU/I	0.9147	0.9734	0.8229
	Triglycerides	91.5 ± 42.11 (6)	65.5 ± 31.51 (4)	67.25 ± 20.6 (4)	128 ± 167.3 (4)	10–100	mg/dL	0.5832	0.8017	0.2828
Electrolytes	Na^+^	150.7 ± 2.43 (7)	150.3 ± 3.15 (7)	150.4 ± 5.86 (5)	147.3 ± 8.22 (4)	145–151	mmol/L	0.9993	0.9972	0.6714
	K^+^	4.94 ± 0.53 (7)	4.76 ± 0.38 (7)	4.67 ± 0.67 (5)	4.93 ± 0.25 (4)	3.9–5.1	mmol/L	0.4296	0.1949	0.9979
	Ca^2+^	1.30 ± 0.05 (7)	1.27 ± 0.09 (7)	1.27 ± 0.12 (5)	1.23 ± 0.18 (4)	1.16–1.40	mmol/L	0.9998	0.9986	0.4783
	Cl^−^	115.1 ± 4.49 (7)	116.7 ± 6.75 (7)	117.4 ± 5.13 (5)	111.5 ± 111.5 (4)	110–119	mmol/L	0.8779	0.4756	0.8899

## Data Availability

Data will be made available upon request.

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
