# Peer review of "Safety of Gonadal Tissue-Derived Mesenchymal Stem Cell Therapy in Geriatric Dogs with Chronic Disease"

_animals, 2024, doi:10.3390/ani14142134_

Round 1
Reviewer 1 Report
Comments and Suggestions for Authors
- abstract/background should clarify which chronic diseases were chosen to be included in the criteria for MSCs treatment and provide associated rationale.
- number of animals (sample size) that received MSCs should be clearly stated in summary/abstract to emphasize safety profile.
- what's the rationale for choosing whether patients would receive allogeneic or autologous cells? Please elaborate.
- which randomization method was used to assign whether the patients would be in the gonadal-derived MSC group or adipose group?
- It's blatantly obvious to the reviewer that several aspects of the control VS gonadal-derived MSC groups are not comparable (frequency of treatment/age of patients/disease targets etc.) - please further explain why comparability were not attempted.
- aside from safety profile, what's the outcome of the disease being treated? without this information, the manuscript lacks major substance.
Author Response
Author's Reply to the Review Report (Reviewer 1)
COMMENT 1. abstract/background should clarify which chronic diseases were chosen to be included in the criteria for MSCs treatment and provide associated rationale.
RESPONSE: Thanks for your detailed comment. Our clinical cases show that many patients desiring mesenchymal stem cell therapy are either elderly, have long-term managed diseases, or have shown poor response to conventional treatments. Therefore, with the objective of elucidating the safety of gonadal tissue-derived stem cells in comparison to adipose-derived stem cells, the two groups were selected to include patients with similar age and conditions.
We have modified the last part of the summary and materials and methods as follows. We hope our response has been appropriately edited.
In the summary section, p.1, line 12-17
“Nineteen geriatric dogs (Age, 8-20 years) with chronic diseases, including myxomatous mitral valve disease, chronic kidney disease, intervertebral disk disease, cognitive dysfunction syndrome, degenerative joint disease, osteoarthritis and progressive retinal atrophy, were participated in a retrospective study. They were divided into two groups: a control group treated with adipose tissue-derived mesenchymal stem cells (MSCs) (n=9) and a treatment group receiving MSCs derived from gonadal tissue (n=10).”
In the materials and methods section, p.2, line 92-94
“In this study, geriatric patients aged 8 to 20 years with degenerative diseases or chronic diseases managed for more than three months were included. The diseases comprised MMVD, CKD, IVDD, CDS, DJD, OA, and PRA.”
COMMENT 2. number of animals (sample size) that received MSCs should be clearly stated in summary/abstract to emphasize safety profile.
RESPONSE: Thanks for your comment. We have modified the last part of the summary and abstract as follows. We hope our response has been appropriately edited.
In the summary section, p.1, line 12-17
“Nineteen geriatric dogs (Age, 8-20 years) with chronic diseases, including myxomatous mitral valve disease, chronic kidney disease, intervertebral disk disease, cognitive dysfunction syndrome, degenerative joint disease, osteoarthritis and progressive retinal atrophy, were participated in a retrospective study. They were divided into two groups: a control group treated with adipose tissue-derived mesenchymal stem cells (MSCs) (n=9) and a treatment group receiving MSCs derived from gonadal tissue (n=10).”
In the abstract section, p.1, line 22-25
“Ensuring the safety of mesenchymal stem cell (MSC) therapy is a fundamental requirement in clinical practice. This study aimed to assess the safety of using gonadal tissue-derived MSCs (n=10) compared to the commonly utilized adipose tissue-derived MSCs (n=9) in geriatric dogs with chronic diseases.”
COMMENT 3. what's the rationale for choosing whether patients would receive allogeneic or autologous cells? Please elaborate.
RESPONSE: Thanks for your detailed comments. The stem cells used in this study were differentiated into stem cells from gonadal tissue discarded during neutering surgery at a veterinary hospital, as well as adipose tissue collected during neutering surgery. As a result, allogeneic stem cells were injected into geriatric dogs. According to previous studies, injection of allogeneic stem cells into dogs has been widely reported (Reference 1). Some studies comparing cell potency based on donor age have shown that stem cells harvested from older donors exhibit significantly reduced potency compared to those harvested from younger donors (References 2, 3). Given that this study targeted diseased dogs aged 8 years and older, all subjects underwent allogeneic MSC therapy. This is described in the discussion section as follows.
In the discussion section, p.13, line 381-387
“In this study, mesenchymal stem cells were derived from fat or gonadal tissue discarded during neutering surgery in dogs under one year of age at a veterinary hospital. According to previous studies, the injection of allogeneic stem cells into dogs has been widely reported [23]. Notably, studies comparing cell potency based on donor age have shown that stem cells harvested from older donors exhibit significantly reduced potency compared to those from younger donors [24,25]. Consequently, for this study, which targeted diseased dogs aged 8 years and older, all patients received allogeneic MSC therapy.”
Reference 1: Kang MH, Park HM. Challenges of stem cell therapies in companion animal practice. J Vet Sci. 2020 Apr;21(3):e42. https://doi.org/10.4142/jvs.2020.21.e42
Reference 2: Lee J, Lee KS, Kim CL, Byeon JS, Gu NY, Cho IS, Cha SH. Effect of donor age on the proliferation and multipotency of canine adipose-derived mesenchymal stem cells. J Vet Sci. 2017 Jun 30;18(2):141-148. doi: 10.4142/jvs.2017.18.2.141. PMID: 27456768; PMCID: PMC5489460.
Reference 3: Taguchi T, Borjesson DL, Osmond C, Griffon DJ. Influence of Donor's Age on Immunomodulatory Properties of Canine Adipose Tissue-Derived Mesenchymal Stem Cells. Stem Cells Dev. 2019 Dec 1;28(23):1562-1571. doi: 10.1089/scd.2019.0118. Epub 2019 Nov 11. PMID: 31588862.
COMMENT 4. which randomization method was used to assign whether the patients would be in the gonadal-derived MSC group or adipose group?
RESPONSE: Thanks for your detailed comments. At VIP Animal Hospital, mesenchymal stem cells extracted from fat tissue and gonad tissue are prepared, and the origin and side effects of the mesenchymal stem cell tissue are fully explained to the animal owner before being injected into the animal. As a result, a total of 69 patients received mesenchymal stem cells during the data collection period (November 21, 2020, to March 4, 2023), and a total of 19 dogs that were elderly and had chronic diseases met the criteria, and a retrospective study and analysis was conducted based on this. In this regard, it is described in the Material and Method section as follows. We hope that these modifications will help adapt our findings to your journal.
In the materials and methods section, p.2, line 74-83
“All medical records of patients who treated with MSCs were reviewed retrospectively. Mesenchymal stem cells extracted from fat and gonad tissues were prepared for injection into animals, with the origin and potential side effects of the MSCs fully explained to the animal owners beforehand. During the data collection period from November 21, 2020, to March 4, 2023, a total of 69 patients received MSC therapy. Among these, 19 elderly dogs with chronic diseases met the criteria for a retrospective study and analysis. In this study, all medical records of patients treated with MSCs at the VIP Animal Medical Center during the specified period were reviewed retrospectively. The review procedure was approved by the Institutional Animal Care and Use Committee (IACUC) of VIP Animal Medical Center (Protocol no. VIP-0006-SC).”
COMMENT 5. It’s blatantly obvious to the reviewer that several aspects of the control VS gonadal-derived MSC groups are not comparable (frequency of treatment/age of patients/disease targets etc.) – please further explain why comparability were not attempted.
RESPONSE: Thanks for your comments. As companion animal grow older, the application of stem cells to elderly dogs is expected to gradually increase. However, no previous studies have been conducted on the safety of stem cells in these aged dogs. The first goal of this study was to confirm the safety of gonadal stem cells in elderly animals, and the second was to confirm that stem cell administration is safe even in elderly animals with chronic diseases, as most of these elderly animals have chronic diseases. Of course, as you said, this study was retrospective, making it difficult to perfectly control the conditions of the patient group and the control group. Nevertheless, this study is the result of evaluating the safety of gonadal stem cells applied to elderly dogs with various chronic diseases, and is important evidence for future application of gonadal stem cells to elderly companion animals. The limitations of this paper are described in the discussion section as follows.
In the discussion section, p.15, line 466-472
“This study has several limitations. The retrospective design limited our ability to precisely control for patient and control group conditions. Additionally, our capacity to monitor adverse effects was constrained to a one-year post-stem cell therapy period within a small patient group. The broad spectrum of diseases targeted within a limited sample size also introduces constraints regarding disease-specificity. Future prospective studies with larger sample sizes, longer follow-up periods, and more stringent controls are necessary to further validate our findings and address these limitations.”
COMMENT 6. aside from safety profile, what's the outcome of the disease being treated? without this information, the manuscript lacks major substance.
RESPONSE: Thanks for your detail comments. The overarching goal of this study was to evaluate the safety of gonadal stem cells in older dogs. It was confirmed that these elderly dogs were accompanied by various chronic diseases, and it was confirmed that the dogs enrolled in this study had no life-threatening adverse effects regardless of the accompanying chronic diseases. Prognosis for comorbidities was assessed in dogs administered stem cells, but this is supplementary information and not the main information of this study. Supplementary data related to the patient's treatment results was attached through scoring. In this regard reference has been made to the following parts of the text. We hope that these revisions will help prepare our research for publication in your journal.
In materials and methods section, p.6, line 233-236
“Patient outcomes were evaluated according to the criteria established in Shah K's study (Reference 1). The treatment response was recorded as ‘Substantial’ (1; excellent improvement), ‘Positive’ (2; good improvement), ‘No change’ (3; no improvement), or ‘Worse’ (4; deterioration in the patients’ condition due to MSCs therapy).”
In the results section, p.13, line 339-342
“3.7. Therapeutic outcomes of MSCs therapy
The patients were scored by veterinarians based on the criteria for stem cell therapy outcomes. There were 11 cases of excellent improvement, 5 cases of good improvement, 3 cases of no improvement (Supplementary 5).”

Reference 1: Shah K, Drury T, Roic I, Hansen P, Malin M, Boyd R, Sumer H, Ferguson R. Outcome of Allogeneic Adult Stem Cell Therapy in Dogs Suffering from Osteoarthritis and Other Joint Defects. Stem Cells Int. 2018 Jun 28;2018:7309201. doi: 10.1155/2018/7309201. PMID: 30050578; PMCID: PMC6046133.
Reviewer 2 Report
Comments and Suggestions for Authors
The presented manuscript descripts the retrospective studies on safety parameters in several elderly canine patients that received allogeneic gonadal tissue-derived mesenchymal stem cells (GT-MSC) for various disease conditions, in comparison to the patients that received allogeneic adipose tissue-derived MSCs (AT-MSC). Authors analyzed data from 19 patients in total and showed, that the systemic administration of allogeneic GT-MSCs are as safe as administrations of allogeneic AT-MSCs. Study concerns the promising and intensively developing veterinary regenerative medicine.
Although the aim of this study was clearly presented, there are multiple questions regarding the study design and tested object.
For study design, there are multiple questions regarding the patient inclusion criteria, treated diseases and criteria for selection of therapy parameters enclosed in the manuscript.
The most important from my point of view is to determine if the cellular products that were assessed retrospectively were standardized, which is crucial for comparison between patients.
Multiple publications emphasized the importance of standardization of MSC products for proper evaluation of their efficacy and safety, as for biological medicinal products the final composition and activity of the product highly depends on the manufacturing method and implemented quality control.1-3 Without it, any conclusions from such clinical trials are easy to undermine by the lack of product specification and repeatability in manufacturing. Also, manufacturing of any parenteral medicinal product without proper safety measures brings a significant risk to the patient. Moreover, non-authorized medicinal products or therapies cannot be offered without scientific proof that they are safe and efficient. Therefore, the value of such clinical studies is very limited.
Specific comments and questions were added to the manuscript pdf as comments.
References
1. Guest DJ et al. (2022) Position Statement: Minimal Criteria for Reporting Veterinary and Animal Medicine Research for Mesenchymal Stromal/Stem Cells in Orthopedic Applications. Front. Vet. Sci., Volume 9, https://doi.org/10.3389/fvets.2022.817041.
2. Luo H et al. (2021) Manufacturing and banking canine adipose-derived mesenchymal stem cells for veterinary clinical application. BMC Vet Res. 17:96
3. Ivanovska A et al. (2022) Manufacturing Mesenchymal Stromal Cells for the Treatment of Osteoarthritis in Canine Patients: Challenges and Recommendations. Front. Vet. Sci, Volume 9, https://doi.org/10.3389/fvets.2022.897150

Author Response
Author's Reply to the Review Report (Reviewer 2)
COMMENT 1. The presented manuscript descripts the retrospective studies on safety parameters in several elderly canine patients that received allogeneic gonadal tissue-derived mesenchymal stem cells (GT-MSC) for various disease conditions, in comparison to the patients that received allogeneic adipose tissue-derived MSCs (AT-MSC). Authors analyzed data from 19 patients in total and showed, that the systemic administration of allogeneic GT-MSCs are as safe as administrations of allogeneic AT-MSCs. Study concerns the promising and intensively developing veterinary regenerative medicine.
Although the aim of this study was clearly presented, there are multiple questions regarding the study design and tested object.
For study design, there are multiple questions regarding the patient inclusion criteria, treated diseases and criteria for selection of therapy parameters enclosed in the manuscript.
The most important from my point of view is to determine if the cellular products that were assessed retrospectively were standardized, which is crucial for comparison between patients.
Multiple publications emphasized the importance of standardization of MSC products for proper evaluation of their efficacy and safety, as for biological medicinal products the final composition and activity of the product highly depends on the manufacturing method and implemented quality control.1-3 Without it, any conclusions from such clinical trials are easy to undermine by the lack of product specification and repeatability in manufacturing. Also, manufacturing of any parenteral medicinal product without proper safety measures brings a significant risk to the patient. Moreover, non-authorized medicinal products or therapies cannot be offered without scientific proof that they are safe and efficient. Therefore, the value of such clinical studies is very limited.
Specific comments and questions were added to the manuscript pdf as comments.
References
- Guest DJ et al. (2022) Position Statement: Minimal Criteria for Reporting Veterinary and Animal Medicine Research for Mesenchymal Stromal/Stem Cells in Orthopedic Applications. Front. Vet. Sci., Volume 9, https://doi.org/10.3389/fvets.2022.817041.
- Luo H et al. (2021) Manufacturing and banking canine adipose-derived mesenchymal stem cells for veterinary clinical application. BMC Vet Res. 17:96
- Ivanovska A et al. (2022) Manufacturing Mesenchymal Stromal Cells for the Treatment of Osteoarthritis in Canine Patients: Challenges and Recommendations. Front. Vet. Sci, Volume 9, https://doi.org/10.3389/fvets.2022.897150
RESPONSE: Thanks for your detailed comments. As you mentioned, we think it's very important to establish standardization when administering stem cells to companion animals. Various conditions for standardizing stem cell administration have been reported in previous papers (Reference 1-6). Based on these papers, we are applying stem cell treatment to patients by standardizing the age, passage, characteristics of tissue donors, and the number of stem cell administrations to patients In addition, we established our own quality control tests and stem cell therapy protocols and adhered to them strictly. This is described in the discussion section as follows. We hope that our modifications are appropriate.
In the discussion section, p.16, line 458-465
“As a minimum effort to ensure safe and effective treatment, we have prepared a precise treatment process. Establishing standardization in the administration of stem cells to companion animals is of paramount importance. Various conditions for standardizing stem cell administration have been documented in previous studies [24,43-47]. Drawing upon these studies, we have implemented standardized protocols for stem cell treatment, including criteria for the age and passage number of stem cells, the characteristics of tissue donors, and the dosage of stem cell administrations. Furthermore, we have developed and strictly adhered to our own quality control tests and stem cell therapy protocols.”
Reference 1: Lee J, Lee KS, Kim CL, Byeon JS, Gu NY, Cho IS, Cha SH. Effect of donor age on the proliferation and multipotency of canine adipose-derived mesenchymal stem cells. J Vet Sci. 2017 Jun 30;18(2):141-148. doi: 10.4142/jvs.2017.18.2.141. PMID: 27456768; PMCID: PMC5489460.
Reference 2: Lee KS, Kang HW, Lee HT, Kim HJ, Kim CL, Song JY, Lee KW, Cha SH. Sequential sub-passage decreases the differentiation potential of canine adipose-derived mesenchymal stem cells. Res Vet Sci. 2014 Apr;96(2):267-75. doi: 10.1016/j.rvsc.2013.12.011. Epub 2013 Dec 30. PMID: 24447790.
Reference 3: Dominici M, Le Blanc K, Mueller I, Slaper-Cortenbach I, Marini F, Krause D, Deans R, Keating A, Prockop Dj, Horwitz E. Minimal criteria for defining multipotent mesenchymal stromal cells. The International Society for Cellular Therapy position statement. Cytotherapy. 2006;8(4):315-7. doi: 10.1080/14653240600855905. PMID: 16923606.
Reference 4: Kabat M, Bobkov I, Kumar S, Grumet M. Trends in mesenchymal stem cell clinical trials 2004-2018: Is efficacy optimal in a narrow dose range? Stem Cells Transl Med. 2020 Jan;9(1):17-27. doi: 10.1002/sctm.19-0202. Epub 2019 Dec 5. PMID: 31804767; PMCID: PMC6954709.
Reference 5: Gugjoo, M. B., Pal, A., & Sharma, G. T. (2020). Dog mesenchymal stem cell basic research and potential applications. Mesenchymal Stem Cell in Veterinary Sciences. Springer Nature Singapore Pte Ltd, 213-282.
Reference 6: Baranovskii DS, Klabukov ID, Arguchinskaya NV, Yakimova AO, Kisel AA, Yatsenko EM, Ivanov SA, Shegay PV, Kaprin AD. Adverse events, side effects and complications in mesenchymal stromal cell-based therapies. Stem Cell Investig. 2022 Nov 8;9:7. doi: 10.21037/sci-2022-025. PMID: 36393919; PMCID: PMC9659480.
COMMENT 2. p.1, line11
As authors declare that it is a clinical study, it would be convenient for the reader to read in this simple summary what was the target disease, number of animals, groups, type of the study (controlled/blinded/randomized or not), safety parameters that were evaluated, general outcome
RESPONSE: Thanks for your detailed comments. This is described in the simple summary section as follows.
In the summary section, p.1, line 12-21
“Simple Summary: Nineteen geriatric dogs (Age, 8-20 years) with chronic diseases, including myxomatous mitral valve disease, chronic kidney disease, intervertebral disk disease, cognitive dysfunction syndrome, degenerative joint disease, osteoarthritis and progressive retinal atrophy, were participated in a retrospective study. They were divided into two groups: a control group treated with adipose tissue-derived mesenchymal stem cells (MSCs) (n=9) and a treatment group receiving MSCs derived from gonadal tissue (n=10). MSCs therapies were administered intravenously at monthly intervals, with multiple frequency using allogeneic MSCs. Safety was evaluated through short-term and long-term physical exams, blood tests, imaging studies, and monitoring of adverse events. No clinical adverse effects were observed in the dogs treated with gonadal tissue-derived MSCs.”
COMMENT 3. p.1, line 13
In which tradition? I think there is no tradition in this field, but only few authorized cell-based medicinal products and many many non-authorized therapies
RESPONSE: I apologize for the inappropriate choice of words. The sentence in question has been removed.
COMMENT 4. p.1, line 14
Which one?
RESPONSE: Thanks for your detailed comments. To elucidate the safety of gonadal tissue-derived mesenchymal stem cells, a comparison was made with adipose tissue-derived MSCs, which are commonly used in conventional mesenchymal stem cell therapy. Patients with similar ages and conditions were selected for this study. The study included patients aged 8 to 20 years with degenerative diseases or chronic conditions managed for more than three months. The conditions included myxomatous mitral valve disease (MMVD), chronic kidney disease (CKD), intervertebral disk disease (IVDD), cognitive dysfunction syndrome (CDS), degenerative joint disease (DJD), osteoarthritis (OA), and progressive retinal atrophy (PRA).
This is described in the materials and methods section as follows. We hope our response has been appropriately edited.
In the materials and methods, p.3, line 92-94
“In this study, patients aged 8 to 20 years with degenerative diseases or chronic conditions managed for more than three months were included. The conditions comprised MMVD, CKD, IVDD, CDS, DJD, OA, and PRA.”
COMMENT 5. p.1, line 14
How many?, Which route?
RESPONSE: Thanks for your detailed comments. All 19 dogs received intravenous stem cell therapy. This is described in the simple summary section as follows. We hope our response has been appropriately edited.
In the simple summary section, p.1, line 15-18
“They were divided into two groups: a control group treated with adipose tissue-derived mesenchymal stem cells (MSCs) (n=9) and a treatment group receiving MSCs derived from gonadal tissue (n=10). MSCs therapies were administered intravenously at monthly intervals, with multiple frequency using allogeneic MSCs.”
COMMENT 6. p.2, line 60-61
Despite such statement, Authors still decided to perform an allogeneic treatment in all investigated individuals, according to table 3.
How the risk of side effects was managed as the investigated population were client-owned dogs?
RESPONSE: Thanks for your detailed comments. Information regarding the potential adverse effects of stem cell therapy was provided in writing, and the guardian provided consent. This information was included in the supplementary 4. The following details have been added to the materials and methods section. We hope that our revisions have been appropriately addressed.
In the materials and methods section, p.5, line 195-198
“The guardians were provided with written information regarding the potential adverse effects and complications of stem cell therapy, and they provided their consent. The consent form is included in the supplementary materials (Supplementary 4).”

COMMENT 7. p.2, line 70
It is unclear for me what analyzed population received. I understood that standard treatment offered by VIP Center is based on adipose-derived MSC, and herein the Authors describe the experimental treatment with gonadal tissue-derived MSCs. However, the study was carried out as retrospective analysis, therefore I deduce that the application of gonadal-MSC was performed before any safety assessment - is that right?
RESPONSE: Thanks for your detailed comments. Stem cells can be derived from various tissues. Numerous studies have reported the application of stem cells derived from different sources such as fat, synovial membranes, tendon, muscle, umbilical cord, and bone marrow to patients (Reference 1). Currently, the VIP treatment system for supplying stem cells to patients is well established, and stem cells from various tissues, not just fat, are being provided to patients. However, previous studies have not reported treatments using stem cells derived from gonadal tissue in elderly patients. Therefore, a retrospective analysis was conducted to confirm the safety of such treatments. This is detailed in the discussion section. We hope that our response and the revised sections have been appropriately adjusted.
Reference 1: Via AG, Frizziero A, Oliva F. Biological properties of mesenchymal Stem Cells from different sources. Muscles Ligaments Tendons J. 2012 Oct 16;2(3):154-62. PMID: 23738292; PMCID: PMC3666517.
“To date, no large-scale clinical studies involving these MSCs in actual patient populations have been reported. Therefore, our aim is to validate the suitability of gonadal tissue as a source of MSCs by assessing the safety of treatment in elderly canine patients.
In this study, mesenchymal stem cells were derived from fat or gonadal tissue discarded during neutering surgery in dogs under one year of age at a veterinary hospital. According to previous studies, the injection of allogeneic stem cells into dogs has been widely re-ported [23]. Notably, studies comparing cell potency based on donor age have shown that stem cells harvested from older donors exhibit significantly reduced potency compared to those from younger donors [24, 25]. Consequently, for this study, which targeted diseased dogs aged 8 years and older, all patients received allogeneic MSC therapy.
The VIP treatment system for supplying stem cells to patients is currently well established, providing stem cells from a variety of tissues, not limited to fat. Therefore, a retrospective analysis was conducted to confirm the safety of such treatments.
COMMENT 8. p.2, line 71-72
This is a very general description, what particular disease conditions were treated? Was it authorised medicine or?
RESPONSE: Thanks for your detailed comments. The primary objective of this study was to evaluate the safety of stem cell administration in elderly patients. This study does not allow for definitive conclusions regarding the therapeutic effects on specific diseases, highlighting the need for further research. While the administration of stem cells was intended to treat certain conditions, it was often used as a supportive therapy.
In the discussion section, p.16, line 469-472
“The broad spectrum of diseases targeted within a limited sample size also introduces constraints regarding disease-specificity. Future prospective studies with larger sample sizes, longer follow-up periods, and more stringent controls are necessary to further validate our findings and address these limitations.”
Reference 1: Hoang DM, Pham PT, Bach TQ, Ngo ATL, Nguyen QT, Phan TTK, Nguyen GH, Le PTT, Hoang VT, Forsyth NR, Heke M, Nguyen LT. Stem cell-based therapy for human diseases. Signal Transduct Target Ther. 2022 Aug 6;7(1):272. doi: 10.1038/s41392-022-01134-4. PMID: 35933430; PMCID: PMC9357075.
COMMENT 9. p. 2, line 79
Could Authors specify which tests were employed to exclude cancer?
RESPONSE: Thanks for your detailed comments. We conducted VDI pre-stem cell panel (with cancer panel), blood test, and chest and abdominal imaging evaluations.
In the materials and methods section, p.3, line 103-104
“For tumor evaluation, physical examinations, blood tests, and imaging examinations were conducted annually.”
COMMENT 10. p.2, line 81-82
This sentence suggests that Authors selected particular medical cases for this retrospective analysis. Excluding the patients with worse health status could have an influence on the conclusions regarding safety of the therapy.
I suggest to enclose the whole list of patients and provide precise inclusion and exclusion criteria, giving also the information how many dogs were on the list and how many were excluded from analysis and why
RESPONSE: Thanks for your detailed comments.
At your request, the complete list of participants in the study has been included (Supplementary 4). We selected patients with similar age, disease, sex, and weight. We did not administer stem cell therapy to patients with severe conditions such as tumors, cardiogenic pulmonary edema, sepsis, thromboembolism, or infectious diseases. Patients who could not undergo short-term or long-term follow-up after stem cell therapy or who were missing main evaluation items from this study were excluded. For these reasons, out of the initial 69 dogs, 50 were excluded, leaving a final total of 19 dogs.
In the materials and methods section, p.3, line 85-91
“Patients were selected based on similar age, disease, sex, and weight parameters. Stem cell therapy was excluded for patients with severe conditions such as tumors, cardiogenic pulmonary edema, sepsis, thromboembolism, or infectious diseases. Furthermore, patients who were unable to undergo short-term or long-term follow-up post-stem cell therapy or who lacked essential evaluation metrics required for this study were excluded. As a result, out of the initial 69 dogs, 50 were excluded, yielding a final sample size of 19 dogs.”

COMMENT 11. p.2, line 83-85.
Why such approach? What is the criterion for repeated session? If the therapy works quickly - patient does not require another session, thus it has small session number? And in contrast, patients that do not respond very well have repeated sessions?
RESPONSE: Thanks for your detailed comments. The number of treatments varied based on the patient's clinical symptoms, the extent of disease improvement, the duration of symptom relief, and the owner's financial capacity. Safety studies on multiple treatment sessions have been conducted for various diseases, and systemic side effects are not known to occur (Reference 1). Some studies have reported positive effects of increasing the number of treatment sessions (Reference 2,3). In the case of intravenous injections, significant cell trapping occurs in the liver and lungs, reducing the number of remaining stem cells (Reference 4). To provide sufficient treatment, multiple sessions were sometimes conducted, resulting in high satisfaction from owners, leading to consistent treatments.
Reference 1: Vij R, Prossin A, Tripathy M, Kim H, Park H, Cheng T, Lotfi D, Chang D. Long-term, repeated doses of intravenous autologous mesenchymal stem cells for a patient with Parkinson's disease: a case report. Front Neurol. 2023 Sep 27;14:1257080. doi: 10.3389/fneur.2023.1257080. PMID: 37840944; PMCID: PMC10569690.
Reference 2: Vij R, Prossin A, Tripathy M, Kim H, Park H, Cheng T, Lotfi D, Chang D. Long-term, repeated doses of intravenous autologous mesenchymal stem cells for a patient with Parkinson's disease: a case report. Front Neurol. 2023 Sep 27;14:1257080. doi: 10.3389/fneur.2023.1257080. PMID: 37840944; PMCID: PMC10569690.
Reference 3: Gong, C., Chang, L., Sun, X. et al. Infusion of two-dose mesenchymal stem cells is more effective than a single dose in a dilated cardiomyopathy rat model by upregulating indoleamine 2,3-dioxygenase expression. Stem Cell Res Ther 13, 409 (2022). https://doi.org/10.1186/s13287-022-03101-w
Reference 4: Oh MS, Lee SG, Lee GH, Kim CY, Kim EY, Song JH, Yu BY, Chung HM. In vivo tracking of 14C thymidine labeled mesenchymal stem cells using ultra-sensitive accelerator mass spectrometry. Sci Rep. 2021 Jan 14;11(1):1360. doi: 10.1038/s41598-020-80416-9. PMID: 33446731; PMCID: PMC7809063.
COMMENT 12. p.2, line 90-92. “They were thoroughly screened for infectious diseases and demonstrated no abnormal findings based on both blood and imaging examinations.”
Was it done in accordance to any guideline or Pharmacopoeia?
RESPONSE: Thanks for your detailed comments. We established our guidelines by following the recommendation checklist presented in the human medical research paper (Reference 1).
Reference 1: Baranovskii DS, Klabukov ID, Arguchinskaya NV, Yakimova AO, Kisel AA, Yatsenko EM, Ivanov SA, Shegay PV, Kaprin AD. Adverse events, side effects and complications in mesenchymal stromal cell-based therapies. Stem Cell Investig. 2022 Nov 8;9:7. doi: 10.21037/sci-2022-025. PMID: 36393919; PMCID: PMC9659480.
COMMENT 13. p.3, line 93
The process was approved by the Institutional Animal Care and Use Committee (IACUC) of VIP Animal Medical Center (Protocol no. VIP-0004-SC)
I believe it is an internal committee of the VIP Animal Medical Center, isn't there a conflict of interest as these procedures were probably for profit? Or?
I see that the Center is offering allogeneic stem cells for veterinary medicine. For various conditions. Are they proof-based medicinal therapies?
RESPONSE: Thanks for your detailed comments. The committee includes members who have no affiliation with the study, particularly adhering to strict criteria set by the South Korean government where more than one-third of the IACUC committee comprises external members. Additionally, every year, the responsible national agency reviews to ensure there are no procedural issues. Stem cell procedures were conducted based on evidence-based medicine, utilizing numerous clinical papers as a foundation.
COMMENT 14. p.3, line 96
Did Authors/Manufacturers manufactured medicinal product in accordance to GMP? If not, how was the contamination control strategy employed? Were any quality control tests performed? Was the microbial contamination assessed?
RESPONSE: Thanks for your detailed comments. Although not a GMP facility, stem cell culture was conducted in research facilities equipped with Clean Room facilities including HEPA filters and biocabinets, with personnel wearing protective clothing. We implemented our own quality control system to manage cell quality and conducted mycoplasma testing to ensure freedom from contaminants.
COMMENT 15. p.3, line 91
Which tissues specifically? Testes and ovaries? Or also adipose tissues?
RESPONSE: Thanks for your detailed comments. This is described in the materials and methods section as follows.
In the materials and methods section, p.3, line 107
“We obtained adipose or gonadal tissues (testes or ovaries) that were discarded during the neutering surgery procedures”
COMMENT 16. p.3, line 100.
It is important to discriminate tissues, as different tissues have different 'impurities', for example connective tissue is different in ovaries, testicles an adipose tissue
RESPONSE: Thanks for your detailed comments. We endeavored to finely separate each tissue during the cell isolation process. Stemness was confirmed through quality control procedures, and microscopic examination during successive passages demonstrated a gradual reduction in impurities such as fibroblasts.
COMMENT 17. p.3, line 113.
Stromal vascular fraction of testes and ovaries? Or adipose tissue?
RESPONSE: Thanks for your detailed comments. The sentence was revised to include all tissues (testes, ovaries, or adipose tissue).
In materials and methods section, p.3, line 116
“Donor derived tissue (testis, ovary, or adipose tissue)”
COMMENT 18. p. 4, line 154
What is the influence of this buffer on MSCs? Mainly nn their viability?
RESPONSE: Thanks for your detailed comments. According to a study evaluating the effectiveness of stem cells based on the buffer, it is recommended to store stem cells in saline, 5% DS, heparin in saline, or Hartmann’s solution at 4°C. These solutions, all FDA-approved, are preferable for maintaining high cell viability and efficacy, provided the storage duration is less than 6 hours and no longer than 12 hours, rather than using phosphate-buffered saline (Reference 1).
Reference 1: Sultana T, Dayem AA, Lee SB, Cho SG, Lee JI. Effects of carrier solutions on the viability and efficacy of canine adipose-derived mesenchymal stem cells. BMC Vet Res. 2022 Jan 7;18(1):26. doi: 10.1186/s12917-021-03120-4. PMID: 34996443; PMCID: PMC8739692.
COMMENT 19. p. 4, line 156
Was the stability studies performed to determine the maximum storage time?
RESPONSE: Thanks for your detailed comments. In feline mesenchymal stem cell research, it was confirmed that cell viability is maintained for up to 48 hours when stored at 4 degrees Celsius (Reference 1).
Reference 1: Arzi B, Peralta S, Fiani N, Vapniarsky N, Taechangam N, Delatorre U, Clark KC, Walker NJ, Loscar MR, Lommer MJ, Fulton A, Battig J, Borjesson DL. A multicenter experience using adipose-derived mesenchymal stem cell therapy for cats with chronic, non-responsive gingivostomatitis. Stem Cell Res Ther. 2020 Mar 13;11(1):115. doi: 10.1186/s13287-020-01623-9. PMID: 32169089; PMCID: PMC7071622.
COMMENT 20. p. 4, line 174-175.
The table 3 below shows, that only allogeneic therapy was included to this study. What was the choice criterion?
RESPONSE: Thanks for your detailed comments. In our treatment cases, many patients desiring stem cell therapy were elderly or had long-term managed diseases, making it difficult to perform additional surgery or anesthesia to obtain autologous stem cells. Therefore, we conducted treatments using allogeneic stem cells.
COMMENT 21. p. 4, line 175-176.
This is very interesting sentence. What were the criteria of determination of mentioned parameters? It is important to enclose it as these parameters can be of critical role in both efficacy and safety
RESPONSE: Thanks for your detailed comments.
We evaluated the feasibility of autologous cell therapy based on age, patient health status, and disease. For patients aged 8 years and older, who were the subjects of this study, allogeneic cell therapy was prioritized. Regarding the dosage, it has been reported that therapeutic effects decrease if the dose is significantly above or below a certain level in humans (Reference 1). Based on this concept, we investigated canine clinical studies to determine the median cell dose and applied this to patient treatments (Reference 2).
In this study, intravenous (IV) administration was chosen for elderly patients to achieve systemic therapeutic effects. However, local injections are sometimes preferred for targeting specific diseases. For example, it has been found that IV administration is less effective than local injection for joint diseases (Reference 3). Additionally, it is known that the homing effect to target organs can be reduced in cases where there are lesions in the lungs and liver during IV administration (Reference 4).
At our institution, we administer p4 cells to patients, and it is known that as cell passages progress, the differentiation capacity, surface marker expression, and proliferation ability of the cells decrease (Reference 5). We recommended multiple administrations to the owners based on a study that showed higher therapeutic effects with a two-dose regimen compared to a single dose in heart disease (Reference 6).
Reference 1: Kabat M, Bobkov I, Kumar S, Grumet M. Trends in mesenchymal stem cell clinical trials 2004-2018: Is efficacy optimal in a narrow dose range? Stem Cells Transl Med. 2020 Jan;9(1):17-27. doi: 10.1002/sctm.19-0202. Epub 2019 Dec 5. PMID: 31804767; PMCID: PMC6954709.
Reference 2: Gugjoo, M. B., Pal, A., & Sharma, G. T. (2020). Dog mesenchymal stem cell basic research and potential applications. Mesenchymal Stem Cell in Veterinary Sciences. Springer Nature Singapore Pte Ltd, 213-282.
Reference 3: Mostafa A, Korayem HE, Fekry E, Hosny S. The Effect of Intra-articular versus Intravenous Injection of Mesenchymal Stem Cells on Experimentally-Induced Knee Joint Osteoarthritis. J Microsc Ultrastruct. 2020 May 20;9(1):31-38. doi: 10.4103/JMAU.JMAU_2_20. PMID: 33850710; PMCID: PMC8030543.
Reference 4: Oh MS, Lee SG, Lee GH, Kim CY, Kim EY, Song JH, Yu BY, Chung HM. In vivo tracking of 14C thymidine labeled mesenchymal stem cells using ultra-sensitive accelerator mass spectrometry. Sci Rep. 2021 Jan 14;11(1):1360. doi: 10.1038/s41598-020-80416-9. PMID: 33446731; PMCID: PMC7809063.
Reference 5: Lee KS, Kang HW, Lee HT, Kim HJ, Kim CL, Song JY, Lee KW, Cha SH. Sequential sub-passage decreases the differentiation potential of canine adipose-derived mesenchymal stem cells. Res Vet Sci. 2014 Apr;96(2):267-75. doi: 10.1016/j.rvsc.2013.12.011. Epub 2013 Dec 30. PMID: 24447790.
Reference 6: Gong, C., Chang, L., Sun, X. et al. Infusion of two-dose mesenchymal stem cells is more effective than a single dose in a dilated cardiomyopathy rat model by upregulating indoleamine 2,3-dioxygenase expression. Stem Cell Res Ther 13, 409 (2022). https://doi.org/10.1186/s13287-022-03101-w
COMMENT 22. p. 4, line 182-184.
“Before allogeneic stem cells therapy, Chlorpheniramine maleate 0.2 mg/kg, SC, once
(Histamine, SAMU MEDIAN Co., Ltd., Seoul, Korea) were administered 30 minutes before treatment to prevent immune reactions.”
Did Authors verified potential influence of this substance on MSC? It is widely known that the MSC are immunomodulatory and their activity relies on the status of inflammatory cells of the patient. Therefore there is a potential influence of such premedication on the therapy outcome
RESPONSE: Thanks for your detailed comments. In order to mitigate potential hypersensitivity reactions following allogeneic mesenchymal stem cell therapy, we implemented antihistamine treatment based on established protocols from the previous studies (Reference 1, 2).
Reference 1: Rhew SY, Park SM, Li Q, An JH, Chae HK, Lee JH, Ahn JO, Song WJ, Youn HY. Efficacy and safety of allogenic canine adipose tissue-derived mesenchymal stem cell therapy for insulin-dependent diabetes mellitus in four dogs: A pilot study. J Vet Med Sci. 2021 Apr 9;83(4):592-600. doi: 10.1292/jvms.20-0195. Epub 2021 Feb 8. PMID: 33551441; PMCID: PMC8111340.
Reference 2: Connick P, Kolappan M, Crawley C, Webber DJ, Patani R, Michell AW, Du MQ, Luan SL, Altmann DR, Thompson AJ, Compston A, Scott MA, Miller DH, Chandran S. Autologous mesenchymal stem cells for the treatment of secondary progressive multiple sclerosis: an open-label phase 2a proof-of-concept study. Lancet Neurol. 2012 Feb;11(2):150-6. doi: 10.1016/S1474-4422(11)70305-2. Epub 2012 Jan 10. PMID: 22236384; PMCID: PMC3279697.
COMMENT 23. p. 4, line 185. Were the cells injected directly from syringe to the vein? What was the concentration and viability of cells in the administered fluid at the time of administration?
RESPONSE: Thanks for your detailed comments. An intravenous catheter was placed, and a butterfly needle was connected to slowly administer the solution over 30 minutes while rolling the syringe. The mesenchymal stem cells were diluted in normal saline to a concentration of 106 cells/kg and a volume of 2.5 ml/kg. After staining with methylene blue, live cell counting was performed, confirming that over 99% of the cells were viable.
COMMENT 24. p.5, line 214. Reference to the figure is missing. Also, the figure shows not only gonadal MSCs, but also adipose-derived ones - it is worth mentioning here about the similarities.
RESPONSE: Thanks for your detailed comments. The requested data is attached as supplementary materials.

COMMENT 25. p. 5, line 221-222.This description should be included in the M&M section in study design (two groups gonadal vs adipose? Three groups testies vs ovaries vs adipose?)
RESPONSE: Thanks for your comments. This is described in the materials and methods section as follows.
In the materials and methods section, p.5, line 194-195
“Ten dogs received gonadal tissue-derived MSC therapy in the experimental group, while nine received adipose tissue injections in the control group.”
COMMENT 26. p.6, line 236-238.
Were these therapies based on evidences in accordance to current world standard, which is a target animal safety studies + field GCP clinical trial? Or only basing on scientific publications? It is important to understand, that positive results reported by other groups not necessarily can be reproduced by other group, as the method of product preparation and characterisation for biological products is crucial for efficacy and safety
RESPONSE: Thanks for your comments. We conducted the treatments in accordance with scientific publications (Reference 1). Additionally, we performed characterization test to confirm the mesenchymal stem cell identity and demonstrated the safety in actual patients by showing no abnormalities in both short-term and long-term follow-up examinations.
Reference 1: Gugjoo, M. B., Pal, A., & Sharma, G. T. (2020). Dog mesenchymal stem cell basic research and potential applications. Mesenchymal Stem Cell in Veterinary Sciences. Springer Nature Singapore Pte Ltd, 213-282.
COMMENT 27. p.8, line 273. Authors stated in the abstract, that the patients were evaluated for a year also for any neoplastic changes (cancer), but there is no word how in the M&M section and also here in results section.
RESPONSE: Thanks for your comments. Those are described in the materials and methods and results section as follows.
In the materials and methods section, p.5, line 220-222
“This involved assessing adverse changes in physiological parameters and assessing tumor formation, growth, and any side effects through physical exams, blood tests, and imaging studies.”
In the results section, p.9, line 306-307
“No tumor formation or growth was observed.”
COMMENT 28. p. 9, line 283. Table 7.
Was the differences between baseline and each time point for each group also tested for statistical significance? This could answer if any type of the product had influence on tested parameters.
RESPONSE: Thanks for the detailed comments. The differences between baseline and each time point were analyzed and presented additionally in Table 7, showing no statistical differences.
This is described in the results section as follows.
In the results section, p.9, line 313, table.7
Table 7. Physical parameters of adverse effects after stem cell therapy

COMMENT 29. p.13, line 321. It is true, however any treatment should be standardized to be able to properly assess the safety and efficacy of the medicinal product. There is no information if the analyzed therapies were standardized, or authorized by authorities as the only legal and ethical approach.
RESPONSE: Thanks for your comment. The desire for standardization of stem cell therapy protocols is equally present in both human and veterinary medicine. We have compiled the major protocols suggested in the veterinary clinical studies published so far, which have demonstrated safe and effective treatment, and have strived to follow them. Due to the lack of established safety, efficacy, and standardization akin to medicinal products, we focused on the physical parameters, hematological changes, tumorigenicity, and adverse effects in patients after cell therapy to elucidate the safety of stem cell treatments.
COMMENT 30. p.13, line 365. Please explain for non clinicist
RESPONSE: Thanks for your comment.
In the discussion section, p.14, line 412
The abbreviation DIC was revised to “disseminated intravascular coagulation” in the manuscript.
COMMENT 31. p.14, line 375. “In this study, adipose tissue-derived MSCs, which have been traditionally widely used, were selected as the control group.” I believe in non-standardized, non-GMP, not clinically-proven?
RESPONSE: I apologize for any inappropriate expressions. The word "Traditionally" has been removed from the sentence.
In the discussion section, p.15, line 423
“In this study, adipose tissue-derived MSCs, which have been widely used, were selected as the control group.”
COMMENT 32. p.14, line 375. Was it determined because of the lack of diagnosis within a year or specific tests? The lack of diagnosis does not implicate the lack of neoplasm, as it can develop longer than a year
RESPONSE: Thank you for the valuable advice. Reflecting on your points, I have revised the text to replace the term "cancer" with "neoplasm" and included the phrase "within one year."
In discussion section, p.14, line 432-433
“Additionally, there were no instances of neoplasm occurrence or growth were observed within one year.”
COMMENT 33. p.14, line 406. ...between gonadal and adipose groups?
RESPONSE: Thanks for your detailed comments. After treatment with gonadal tissue-derived stem cells, comparisons of albumin and globulin levels between baseline, short-term, and long-term showed no statistically significant differences. This is described in the discussion section as follows.
In the discussion section, p.16, line 453-456
“The statistical significance of albumin and globulin levels was assessed after gonadal tissue-derived stem cell therapy, but no significant differences were found between baseline, short-term, and long-term measurements.”